# Liver transplant waitlist removal, transplantation rates and post-transplant survival in Hispanics

**Paul J. Thuluvath**[1,2]*, **Waseem Amjad**[1], **Talan Zhang**[1]

1 Institute for Digestive Health & Liver Diseases, Mercy Medical Center, Baltimore, Maryland, United States of America, 2 Department of Medicine, University of Maryland School of Medicine, Baltimore, Maryland, United States of America

* thuluvath@gmail.com

**Data Availability Statement:** The complete datasets are available to all public from the United Network for Organ Sharing. Requests can be made at optn.transplant.hrsa.gov/data/request-data/

## Abstract

### Background and objectives

Hispanics are the fastest growing population in the USA, and our objective was to determine their waitlist mortality rates, liver transplantation (LT) rates and post-LT outcomes.

### Methods

All adults listed for LT with the UNOS from 2002 to 2018 were included. Competing risk analysis was performed to assess the association between ethnic group with waitlist removal due to death/deterioration and transplantation. For sensitivity analysis, Hispanics were matched 1:1 to Non-Hispanics using propensity scores, and outcomes of interest were compared in matched cohort.

### Results

During this period, total of 154,818 patients who listed for liver transplant were involved in this study, of them 23,223 (15%) were Hispanics, 109,653 (71%) were Whites, 13,020 (8%) were Blacks, 6,980 (5%) were Asians and 1,942 (1%) were others. After adjusting for differences in clinical characteristics, compared to Whites, Hispanics had higher waitlist removal due to death or deterioration (adjusted cause-specific Hazard Ratio: 1.034, p = 0.01) and lower transplantation rates (adjusted cause-specific Hazard Ratio: 0.90, p<0.001). If Hispanics received liver transplant, they had better patient and graft survival than Non-Hispanics (p<0.001). Compared to Whites, adjusted hazard ratio for Hispanics were 0.88 (95% CI 0.84, 0.92, p<0.001) for patient survival and 0.90 (95% CI 0.86, 0.94, p<0.001) for graft survival. Our analysis in matched cohort showed the consistent results.

### Conclusions

This study showed that Hispanics had higher probability to be removed from the waitlist due to death, and lower probability to be transplanted, however they had better post-LT outcomes when compared to whites.

using the information outlined in the Methods section.

**Funding:** No funding.

**Competing interests:** No conflict of interest.

## Introduction

There are significant racial disparities in access to the liver transplantation (LT) that includes lower referral rates to LT centers, lower rates of placement on the LT wait list, increased wait-list removal rates and lower LT rates [1–4]. Additionally, blacks are reported to have lower post-liver transplant survival rates when compared to whites [5–8]. The racial disparities in LT have not improved significantly despite the institution of MELD scores for organ allocation or the implementation of Regional Share 35/ National Share 15 to decrease the geographical disparities [4, 9–11].

The US Census Bureau has projected that Hispanic population will almost double by 2060 and they will represent 27.5% of the US population [12]. It has been suggested that Hispanics have a higher life expectancy at birth than non-Hispanic whites [13]. Despite the high life expectancy, the national statistics shows that chronic liver disease is 7th most common cause of death among Hispanics whereas it is not among the top 10 causes of death in non-Hispanic whites and African Americans [14]. The data from the Center for Disease Control (CDC) show that non-Hispanic whites have 30% lower mortality rates due to end stage liver diseases than Hispanic population [15]. As compared to other ethnicities especially Whites, Hispanics have a higher prevalence of nonalcoholic fatty liver disease, worse outcomes with hepatitis C infection, higher incidence of hepatocellular cancer, and heavy episodic alcohol use related liver problems [16–20]. The patients with private insurance are more likely to be referred and listed for the liver transplant and a national inpatient study had shown that the Hispanics are more likely to have Medicaid when compared to Whites [21, 22] This may delay transplant referral and this may result in higher waitlist mortality or removal [21]. There are no comprehensive national studies examining the waitlist removal rates, transplantation rates and post-LT outcomes in Hispanic population. We, therefore, conducted a retrospective cohort analysis of the US national dataset (UNOS data from 2002–2018) to determine racial/ethnic disparities, more specifically among the Hispanics, in waitlist removal rates, liver transplantation rates and post-LT survival outcomes.

## Patients and methods

### Study population

Our retrospective cohort included all adult ($\geq$ 18 years) patients listed for LT with the United Network for Organ Sharing (UNOS) from February 27, 2002 to December 31, 2018. We excluded: 1) those with missing information on the date of listing, 2) who were listed for multiple organ transplantation from this analysis, 3) who had prior liver transplant. Total of 154,818patients were involved in this study, 23,223 (15%) were Hispanics, 109,653 (71%) were Whites, 13,020 (8%) were Blacks, 6,980 (5%) were Asians and 1,942 (1%) were others. The racial groups were defined based on the race/ethnicity recorded at the time of listing with the UNOS.

### Outcomes of interest

For waitlist removal analysis, our outcome of interest was time to removal due to 1) death or deterioration; 2) live transplant. Observation started at an index date, defined as registration date. Patients were followed until the removal from waitlist or were censored at last follow up date. For post-transplant patient and graft survival analysis, time to events included death and graft failure accordingly. The index date was defined as first liver transplant date. Follow-up continued through the earlier date of patient death, graft failure or last follow up date.

### Adjustment for confounding factors

We collected following potential risk factors and confounders at the time of listing and liver transplant including age, gender, body mass index (BMI), obesity (defined as BMI $\geq$ 30), initial waiting list Model for End-stage Liver Disease (MELD) score, Karnofsky Performance Status (KPS scores were not available for the entire study period), presence of diabetes mellitus (DM), hospitalization status (not admitted, admitted to hospital, admitted to intensive care unit), surrogate markers of socioeconomic status (SES) including insurance, education and income status and donor risk index (DRI, in those who had liver transplant). The etiology of liver disease was based on the primary diagnosis at the time of listing. Those with hepatocellular carcinoma (HCC) were grouped as HCC even if they had another primary diagnosis.

Listing region was used for adjustment in all univariate and multivariable analysis.

## Statistical analysis

Descriptive statistics for characteristics of patients were presented as means and standard deviations (SDs) for continuous variables, and frequencies for categorical variables. We did pairwise comparison between different ethnic groups using Bonferroni-adjusted multiple t-tests for continuous variables and Chi square test for categorical variables.

To evaluate the effect of race on waitlist removal due to liver transplant, competing-risks analysis was performed with death as a competing event because the occurrence of one event might preclude the occurrence of the other event. Nonparametric estimate of the cumulative incidence function and Gray's test were used to investigate the difference between ethnic groups on removal due to death and liver transplant. Univariate and multivariable cause-specific Cox models were fitted to assess the association between ethnic group and other covariates with difference causes of removal. Adjusted Cause-specific Hazard Ratios (csHR) based on the fitted model were reported.

To assess and compare overall patient survival and graft survival in different ethnic group, Kaplan-Meier survival curves were used for data illustration. Log-Rank test was used to test if there was significant difference between ethnic groups on patient and graft survival after liver transplant. Cox proportional hazard regressions were used to study the association between the risk factors and patients and graft survival. We started with univariate analysis, followed by multivariable analysis, those with significant effects ($p < 0.05$) were retained in the final model using a backward model selection approach. We used clinical characteristics at the time of liver transplantation for this analysis. For multivariable analysis, we did not exclude any covariates based on univariable models. The final model was selected by balancing goodness-of-fit (e.g., Bayesian information criteria). Estimations of adjusted hazard ratios (aHR) and 95% confidence intervals (CI) for each outcome of interest were reported. Region was used as an adjustment in all univariate and multivariable analysis. All analyses were performed using SAS version 9.4 / STAT 15.1 (SAS Institute Inc., Cary, NC, USA).

For sensitivity analysis, Hispanics were matched 1:1 to Non-Hispanics using propensity scores. The propensity score matching was performed with the R package "MatchIt". We used nearest neighbor matching without replacement, no caliper matching based on the variables: region, age, gender, BMI, albumin, creatinine, type 2 diabetes, encephalopathy, MELD score, KPS, dialysis, portal vein and diagnosis. Balance of index-date covariates between Hispanics and non-Hispanics were compared after matching using Chi square test for categorical variables and t-tests for continuous variables, if data was not normally distributed, nonparametric Wilcoxon test was used. All outcomes of interest were compared between Hispanics and Non-Hispanics in matched cohort.

The study was exempt from institutional review board approval since the data were deidentified.

## Results

The clinical characteristics, etiology of liver disease, liver disease severity, performance status (KPS) and DRI stratified by race at the time of listing are shown in Table 1. There were many differences between the groups with respect to age, sex, BMI, albumin, serum creatinine, DM and MELD scores (Table 1). Compared to Whites, Hispanics were younger (53.7 vs. 54.4 years), had a higher proportion of women (38% vs. 35%), had a higher BMI (29.3 vs. 28.8) and had higher MELD scores (17.9 vs. 17.4) at listing. Severe disability (KPS score less than 40%) was more common in Hispanics (22% vs. 19%) and they were also more likely to be on dialysis (6% vs. 4%). When compared to the whites, Hispanics were more likely to have HCC, HCV, cryptogenic cirrhosis or autoimmune liver disease.

Of the total number of patients (n = 154,818), 56.9% were transplanted, 27.3% died/deteriorated, 9.1% improved or removed for other reasons and 6.6% were still waiting. Among the 42,247 (27.3%) who were removed because of death/deterioration/too sick, only 3,733 (2.4%) were removed because they deteriorated or were too sick. Only 42 patients were inactive at the time of this analysis.

### Waitlist removal due to death or deterioration

One-year and 3-year cumulative incidence (CI) of waitlist removal rates due to death/deterioration was 17% and 29% respectively for Hispanics and 15% and 24% for whites. There were significant differences between ethnic groups on cumulative incidence of waitlist removal due to death or deterioration (Gray's test, p<0.001, Fig 1A). Compared to Whites, Hispanics are more likely to be removed from the list due to death or deterioration (adjusted csHR 1.034, p = 0.01) after adjusting for differences in clinical characteristics and other confounders (Table 2). When Hispanic were compared to others (all Non-Hispanic combined), there was a significantly higher cumulative incidence of waitlist removal rate due to death or deterioration among Hispanics (Gray's test, p <0.001, Fig 1B).

Age (adjusted csHR 1.03, p<0.001), male sex (adjusted csHR 1.08, p<0.001), race, morbid obesity (adjusted csHR 1.15, p<0.001), BMI (p<0.001), presence of diabetes (adjusted csHR 1.09, p<0.001), higher serum albumin (adjusted csHR 0.82, p<0.001), MELD score (adjusted csHR 1.06, p<0.001), portal vein thrombosis (adjusted csHR 0.91, p<0.001), advanced encephalopathy (adjusted csHR 1.52, p<0.001), poor performance status (adjusted csHR 1.56, p<0.001) and causes of liver disease were all significantly associated with waitlist removal due to death/deterioration (Table 2). Listing region was used for adjustment in all univariate and multivariable analysis.

### Waitlist removal due to liver transplantation

During the study period 63,581 (58%) of whites and 11,758 (51%) of Hispanics listed patients had undergone liver transplantation. Of these transplanted patients, 3,803 (6.0%) whites and 694 (5.9%) Hispanics had re-transplantation. The cumulative incidence of transplantation was significantly different between ethnic groups (Gray's test p<0.001, Fig 1A). The cumulative incidence of transplantation in the first 6 months was highest in blacks (40%), followed by whites (36%) and Hispanics (30%). Cumulative incidence in three years was highest in African Americans (60%), followed by whites (56%) and Hispanics (49%). After adjusting for other covariates, compared to Whites, Hispanics (adjusted csHR 0.896, p<0.000) and Blacks (adjusted csHR 0.964, P = 0.003), were less likely to receive liver transplant but Asians

**Table 1. Patients' characteristics and pairwise comparison with Bonferroni-adjusted p value.**

| Variable | White | Black | Hispanic | Asian | Others | Bonferroni-adjusted p value (Compared to White) | | | Hispanics vs. Non-Hispanics |
|---|---|---|---|---|---|---|---|---|---|
| | (N = 109,653) | (N = 13,020) | (N = 23,223) | (N = 6,980) | (N = 1,942) | Black | Hispanic | Asian | |
| Age, Mean (SD) | 54.4 (10.2) | 51.6 (12.2) | 53.7 (10.4) | 54.8 (11.1) | 52.0 (11.0) | <.001 | <.001 | 0.003 | <.001 |
| Female, N (%) | 37941 (35%) | 5558 (43%) | 8805 (38%) | 2306 (33%) | 849 (44%) | <.001 | <.001 | 0.077 | <.001 |
| Albumin, Mean (SD) | 3.1 (0.67) | 2.9 (0.75) | 3.0 (0.68) | 3.3 (0.78) | 3.0 (0.67) | <.001 | <.001 | <.001 | <.001 |
| Serum Creatinine, Mean (SD) | 1.2 (1.00) | 1.5 (1.59) | 1.3 (1.14) | 1.2 (1.19) | 1.3 (1.23) | <.001 | 0.377 | 0.008 | <.001 |
| Dialysis, N (%) | 4203 (4%) | 872 (7%) | 1452 (6%) | 323 (5%) | 124 (6%) | <.001 | <.001 | 0.009 | <.001 |
| Encephalopathy 3–4, N (%) | 7756 (7%) | 1211 (9%) | 1514 (7%) | 463 (7%) | 198 (10%) | <.001 | 0.026 | 1.000 | <.001 |
| BMI, Mean (SD) | 28.8 (5.8) | 28.6 (6.1) | 29.3 (5.7) | 25.2 (4.4) | 29.9 (6.2) | <.001 | <.001 | <.0001 | <.001 |
| Morbid obesity, N (%) | 4507 (4%) | 611 (5%) | 1046 (5%) | 43 (1%) | 125 (6%) | 0.016 | 0.061 | <.001 | <.001 |
| DM type II, N (%) | 19844 (19%) | 2306 (18%) | 5841 (26%) | 1471 (22%) | 457 (24%) | 1.000 | <.001 | <.0001 | <.001 |
| MELD, Mean (SD) | 17.4 (8.9) | 19.6 (10.7) | 17.9 (9.4) | 16.2 (10.6) | 19.1 (9.9) | <.001 | <.001 | <.001 | <.001 |
| PVT, N (%) | 5018 (5%) | 350 (3%) | 1233 (6%) | 252 (4%) | 124 (7%) | <.001 | <.001 | 0.003 | <.001 |
| Poor performance status (KPS 10–40), N (%) | 17127 (19%) | 2586 (24%) | 4241 (22%) | 1158 (20%) | 424 (25%) | <.001 | <.001 | <.001 | <.001 |
| Diagnosis | | | | | | <.001 | <.001 | <.001 | <.001 |
| HCC, N (%) | 19238 (18%) | 2649 (20%) | 4370 (19%) | 2218 (32%) | 351 (18%) | | | | |
| ALD+HCV, N (%) | 5632 (5%) | 615 (5%) | 1426 (6%) | 45 (1%) | 84 (4%) | | | | |
| ALD, N (%) | 19584 (18%) | 823 (6%) | 4021 (17%) | 370 (5%) | 373 (19%) | | | | |
| HCV, N (%) | 23208 (21%) | 3766 (29%) | 5528 (24%) | 945 (14%) | 368 (19%) | | | | |
| HBV, N (%) | 1572 (1%) | 511 (4%) | 262 (1%) | 1570 (22%) | 82 (4%) | | | | |
| AIH, N (%) | 2386 (2%) | 666 (5%) | 743 (3%) | 105 (2%) | 62 (3%) | | | | |
| CC, N (%) | 5897 (5%) | 355 (3%) | 1544 (7%) | 242 (3%) | 81 (4%) | | | | |
| Metabolic, N (%) | 2038 (2%) | 68 (1%) | 161 (1%) | 49 (1%) | 24 (1%) | | | | |
| NASH, N (%) | 10559 (10%) | 211 (2%) | 1883 (8%) | 205 (3%) | 184 (9%) | | | | |
| PBC, N (%) | 3120 (3%) | 271 (2%) | 715 (3%) | 92 (1%) | 64 (3%) | | | | |
| PSC, N (%) | 4563 (4%) | 820 (6%) | 258 (1%) | 107 (2%) | 35 (2%) | | | | |
| Others, N (%) | 11857 (11%) | 2265 (17%) | 2312 (10%) | 1032 (14%) | 295(12%) | | | | |
| Socioeconomic status | | | | | | | | | |
| Highest Education | | | | | | <.001 | <.001 | <.001 | <.001 |
| Grade school (0–8) | 2300 (2%) | 323 (3%) | 4202 (21%) | 691 (12%) | 64 (4%) | | | | |
| High school (9–12) or GED | 42016 (45%) | 5525 (50%) | 10066 (50%) | 1973 (34%) | 858 (49%) | | | | |
| College/Technical school | 24902 (26%) | 2929 (27%) | 3756 (18%) | 1145 (19%) | 496 (28%) | | | | |
| Associate/Bachelor degree | 17882 (19%) | 1578 (14%) | 1751 (9%) | 1301 (22%) | 245 (14%) | | | | |
| Post-college degree | 7262 (8%) | 634 (6%) | 548 (3%) | 777 (13%) | 88 (5%) | | | | |
| Insurance | | | | | | <.001 | <.001 | <.001 | <.001 |
| Private | 66701 (61%) | 6636 (51%) | 10408 (45%) | 4027 (58%) | 906 (47%) | | | | |
| Public-Medicaid | 14202 (13%) | 2765 (21%) | 6419 (28%) | 1475 (21%) | 485 (25%) | | | | |
| Public-Medicare | 23201 (21%) | 2746 (21%) | 5259 (23%) | 1145 (16%) | 378 (19%) | | | | |
| Public-Others | 3568 (3%) | 619 (5%) | 755 (3%) | 111 (2%) | 133 (7%) | | | | |
| Others | 1942 (2%) | 249 (2%) | 371 (2%) | 220 (3%) | 39 (2%) | | | | |
| Work for income | 23041 (25%) | 2633 (24%) | 3454 (17%) | 1792 (30%) | 328 (19%) | 0.059 | <.001 | <.001 | <.001 |

ALD—alcoholic liver disease; AIH—autoimmune hepatitis; BMI—body mass index; CC—cryptogenic cirrhosis; DM—diabetes mellitus; DRI—donor risk index; HCV—hepatitis C; HBV—hepatitis B; HCC—hepatocellular carcinoma; KPS—Karnofsky Performance Status; MELD—model for end-stage liver disease; NASH—nonalcoholic steatohepatitis; PBC—primary biliary cholangitis; PSC—primary sclerosing cholangitis; PVT—portal vein thrombosis; SD—standard deviation.

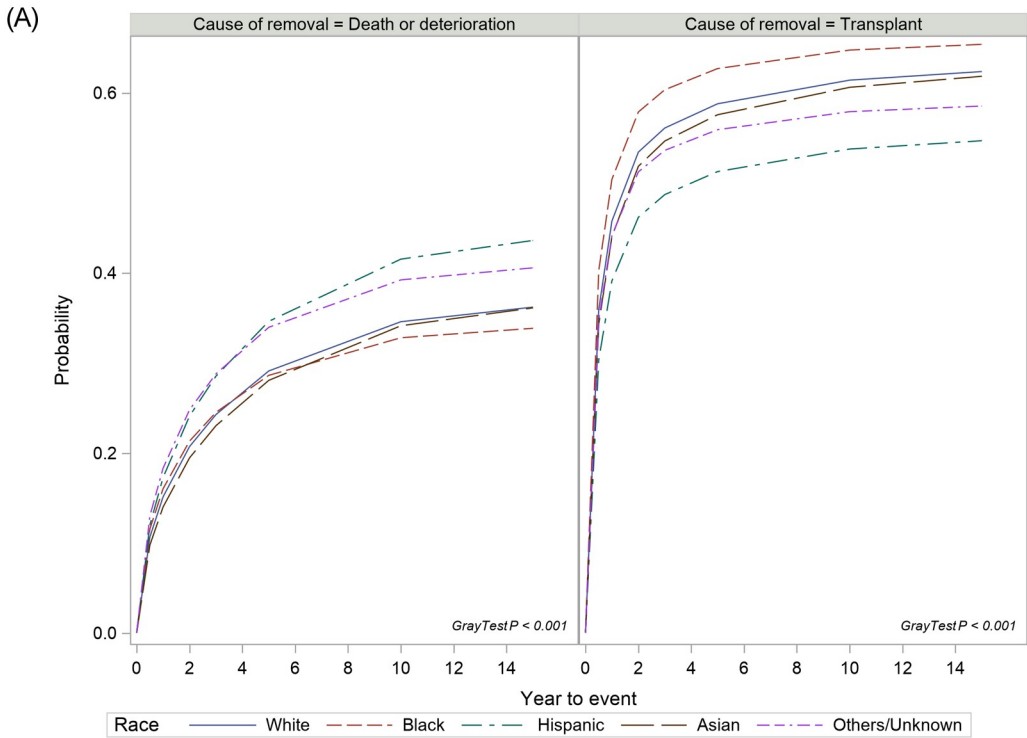

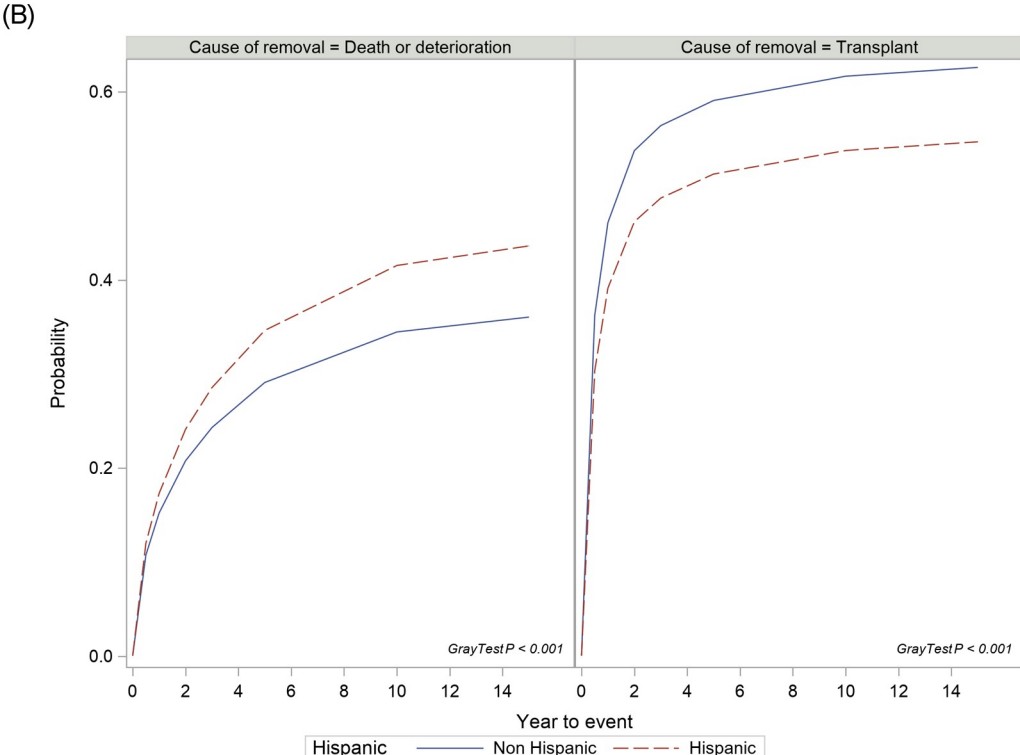

**Fig 1.** A. Cumulative incidence of removal due to death or deterioration and liver transplantation stratified by race; B. Cumulative incidence of removal due to death or deterioration and liver transplantation in Hispanics when compared to other races (combined).

**Table 2. Adjusted cause-specific Hazard Ratios (csHR) on removal due to death or deterioration and transplantation with and without socioeconomic status (SES).**

| | Due to death/deterioration | | | Due to Liver Transplant | | |
|---|---|---|---|---|---|---|
| | Adjusted csHR without SES | Adjusted csHR with SES | P value | Adjusted csHR without SES | Adjusted csHR with SES | P value |
| Race | | | | | | |
| White | 1 | 1 | | 1 | 1 | |
| Black | 1.076 | 1.053 | 0.004 | 0.964 | 0.978 | 0.067 |
| Hispanic | 1.034 | 0.966 | 0.014 | 0.896 | 0.916 | <.0001 |
| Asian | 1.024 | 1.005 | 0.843 | 1.051 | 1.05 | 0.006 |
| Others | 1.188 | 1.152 | 0.000 | 0.99 | 1.007 | 0.831 |
| Age | 1.031 | 1.031 | <.0001 | 0.999 | 1 | 0.676 |
| Gender (Male vs Female) | 1.084 | 1.109 | <.0001 | 1.074 | 1.064 | <.0001 |
| Albumin | 0.819 | 0.825 | <.0001 | 0.927 | 0.925 | <.0001 |
| Encephalopathy (Yes vs No) | 1.521 | 1.524 | <.0001 | 1.162 | 1.165 | <.0001 |
| Morbid Obesity (Yes vs No) | 1.149 | 1.152 | <.0001 | 0.957 | 0.955 | 0.008 |
| DM type 2 (Yes vs. No) | 1.088 | 1.078 | <.0001 | 0.924 | 0.928 | <.0001 |
| MELD score | 1.063 | 1.063 | <.0001 | 1.085 | 1.085 | <.0001 |
| Portal Vein Thrombosis | 0.909 | 0.919 | 0.000 | 0.880 | 0.88 | <.0001 |
| Karnofsky Performance Status | | | | | | |
| 80–100 | 1 | 1 | | 1 | 1 | |
| 50–70 | 1.134 | 1.066 | <.0001 | 1.054 | 1.089 | <.0001 |
| 10–40 | 1.562 | 1.464 | <.0001 | 1.471 | 1.521 | <.0001 |
| Diagnosis | | | | | | |
| Hepatitis C (HCV) | 1 | 1 | | 1 | 1 | |
| Hepatocellular Cancer | 0 | 0 | 0.550 | 3.776 | 3.731 | <.0001 |
| ALD + HCV | 1.114 | 1.081 | <.0001 | 1.098 | 1.107 | <.0001 |
| Alcoholic liver disease | 0.841 | 0.844 | <.0001 | 0.928 | 0.922 | <.0001 |
| Hepatitis B | 0.859 | 0.869 | <.0001 | 1.133 | 1.122 | <.0001 |
| Autoimmune hepatitis | 0.716 | 0.745 | <.0001 | 1.037 | 1.025 | 0.305 |
| Cryptogenic | 0.84 | 0.861 | <.0001 | 1.055 | 1.046 | 0.012 |
| Other metabolic diseases | 0.863 | 0.902 | 0.013 | 1.543 | 1.531 | <.0001 |
| NASH | 0.88 | 0.907 | <.0001 | 1.107 | 1.093 | <.0001 |
| Primary biliary cholangitis | 0.836 | 0.866 | <.0001 | 1.342 | 1.32 | <.0001 |
| PSC | 0.531 | 0.57 | <.0001 | 1.434 | 1.402 | <.0001 |
| Others | 1.052 | 1.073 | <.0001 | 0.937 | 0.932 | <.0001 |
| Recipient highest Education level | | | | | | |
| Grade school (0–8) | | 1 | | | 1 | |
| Hight school (9–12) or GED | | 0.997 | 0.89 | | 1.009 | 0.63 |
| Attended college / technical school | | 0.921 | 0.000 | | 0.997 | 0.87 |
| Associate / Bachelor | | 0.871 | <.0001 | | 1.011 | 0.56 |
| Post-college graduate | | 0.759 | <.0001 | | 1.036 | 0.11 |
| Others/Unknown/Not reported | | 0.988 | 0.610 | | 0.967 | 0.09 |
| Insurance | | | | | | |
| Private | | 1 | | | 1 | |
| Public-Medicaid | | 1.2 | <.0001 | | 0.914 | <.0001 |
| Public-Medicare | | 1.068 | <.0001 | | 0.867 | <.0001 |
| Public-Others | | 1.06 | 0.027 | | 0.92 | <.0001 |
| Others | | 1.301 | <.0001 | | 1.112 | <.0001 |

(*Continued*)

**Table 2.** (Continued)

| | Due to death/deterioration | | | Due to Liver Transplant | | |
|---|---|---|---|---|---|---|
| | Adjusted csHR without SES | Adjusted csHR with SES | P value | Adjusted csHR without SES | Adjusted csHR with SES | P value |
| Employment (Yes vs No) | | 0.882 | <.0001 | | 1.056 | <.0001 |

ALD—alcoholic liver disease; DM—diabetes mellitus; MELLD—model for end-stage liver disease; NASH—nonalcoholic steatohepatitis; PSC—primary sclerosing cholangitis; SD—standard deviation; SES -socioeconomic status.

(adjusted csHR 1.05, p = 0.004) had a higher likelihood of receiving a liver transplant (Table 2). When compared Hispanic with others (all Non-Hispanics combined), Hispanics were less likely to receive liver transplant (Gray's test P<0.001) (Fig 1B).

Age (adjusted csHR 0.99, p<0.001), male sex (adjusted csHR 1.07, p<0.001), race, obesity (adjusted csHR 0.96, p<0.001), presence of diabetes (adjusted csHR 0.92, p<0.001), high serum albumin (adjusted csHR 0.93, p<0.001),, stage 3/4 encephalopathy (adjusted csHR 1.16, p<0.001), MELD score (adjusted csHR 1.09, p<0.001), portal vein thrombosis (adjusted csHR 0.88, p<0.001), poor performance status (adjusted csHR 1.47, p<0.001) and causes of liver disease were significantly associated with removal from the list due to liver transplantation (Table 2).

## Post-transplant patient and graft survival

The clinical characteristics of donors and patients at the time of liver transplantation is shown in Table 3. These variables were used in the multivariable analysis for graft and patient survival.

There were significant differences between ethnic groups on patient survival (p<0.001) and graft survival (p<0.001) (Fig 2A and 2B). Asians had longest mean survival time (12.1 years ± 0.13) followed by Hispanics (11.2 years ± 0.08) and Whites (10.8 years ± 0.04). Patient survival at 10-years was 64% for Hispanics and 60% for whites, and 10-year graft survival was 62% for Hispanics and 58% for whites. Compared to Non-Hispanics, Hispanics had a significant higher survival rate (Fig 2C Log-Rank P<0.001). The risk factors on graft and patient survival were race, age, gender, BMI, MELD score, presence of diabetes, albumin, encephalopathy, portal vein thrombosis, poor performance status and disease etiology (Table 4). More Asians (33%) received relatively poor-quality graft (defined as DRI >2) than whites (28%), blacks (26%) or Hispanics (29%).

After adjusting for other confounders, Hispanics (aHR 0.90, p<0.001) and Asians (aHR 0.71, p<0.0001) had lower risk of post-LT death compared to whites (Table 4). In contrast, blacks had highest risk of death (aHR 1.27, P<0.0001) compared to whites.

## Causes of death after liver transplantation

We classified the cause of death into the following categories: cardiovascular, respiratory failure, graft failure, hemorrhage, malignancy, immunosuppressive drug related, renal failure, operative complications, infection and other/unknown. In our study 16.6% patients died from malignancy, 13.2% from cardiovascular complications, and 12.6% from infection. There were significant differences in the causes of death among the racial groups (p<0.0001). The common causes of death were graft failure (16.3%) in Blacks, cardiovascular complications (12.9%) in Whites or malignancy in Hispanics (15.7%) and Asians (22.4%) (S1 Table).

Table 3. Patient (at the time of transplant) and donor characteristics.

| Variable | White | Black | Hispanic | Asian | Others |
|---|---|---|---|---|---|
| | (N = 63581) | (N = 7847) | (N = 11758) | (N = 3912) | (N = 1035) |
| **Recipients' characteristics at transplantation:** | | | | | |
| Age, Mean (SD) | 54.8 (10.2) | 51.6 (12.2) | 53.8 (10.5) | 55.2 (10.9) | 52.2 (11.0) |
| Recipient gender: Female | 20226 (32%) | 3282 (42%) | 4186 (36%) | 1269 (32%) | 423 (41%) |
| Recipient BMI, Mean (SD) | 28.6 (5.7) | 28.5 (6.2) | 28.9 (5.6) | 25.0 (4.4) | 29.7 (6.0) |
| Morbidly Obese | 2466 (4%) | 378 (5%) | 518 (4%) | 21 (1%) | 68 (7%) |
| Albumin, Mean (SD) | 3.1 (0.7) | 2.9 (0.8) | 3.0 (0.8) | 3.3 (0.8) | 3.0 (0.7) |
| Bilirubin (mg/dL), Mean (SD) | 8.0 (10.5) | 10.0 (11.6) | 10.2 (12.5) | 8.7 (12.5) | 10.1 (12.3) |
| Serum Creatinine, Mean (SD) | 1.4 (1.03) | 1.5 (1.20) | 1.4 (1.10) | 1.2 (0.95) | 1.4 (0.97) |
| INR, Mean (SD) | 1.9 (1.3) | 2.1 (2.1) | 2.0 (1.3) | 1.8 (1.5) | 2.1 (1.2) |
| MELD score, Mean (SD) | 21.3 (10.2) | 22.9 (11.2) | 23.2 (11.4) | 18.9 (12.2) | 23.4 (11.2) |
| Ascites is moderate at transplant | 16883 (27%) | 1754 (22%) | 3256 (28%) | 633 (16%) | 291 (28%) |
| Encephalopathy is 3–4 at transplant | 6593 (10%) | 921 (12%) | 1304 (11%) | 364 (9%) | 156 (15%) |
| KPS categories at transplant | | | | | |
| 10–40 | 17497 (28%) | 2442 (31%) | 4038 (34%) | 953 (24%) | 354 (34%) |
| 50–70 | 22145 (35%) | 2543 (32%) | 3959 (34%) | 1156 (30%) | 361 (35%) |
| 80–100 | 14712 (23%) | 1767 (23%) | 2201 (19%) | 1250 (32%) | 228 (22%) |
| Missing | 9227 (15%) | 1095 (14%) | 1560 (13%) | 553 (14%) | 92 (9%) |
| Child-Pugh categories at transplant | | | | | |
| A:5 or 6 points | 6664 (12%) | 1003 (14%) | 1165 (11%) | 1177 (33%) | 115 (12%) |
| B:7–9 points | 17372 (30%) | 1765 (24%) | 2846 (27%) | 908 (25%) | 215 (22%) |
| C:>9 points | 33780 (58%) | 4449 (62%) | 6705 (63%) | 1481 (42%) | 655 (66%) |
| Medical condition | | | | | |
| In intensive care unit | 6495 (11%) | 1107 (15%) | 1816 (17%) | 557 (16%) | 154 (16%) |
| Hospitalized not in ICU | 9866 (17%) | 1327 (18%) | 2207 (21%) | 465 (13%) | 190 (19%) |
| Not hospitalized | 41448 (72%) | 4783 (66%) | 6685 (62%) | 2545 (71%) | 639 (65%) |
| **Donors' characteristics:** | | | | | |
| Donor age (years), Mean (SD) | 41.9 (16.6) | 40.4 (16.1) | 41.3 (16.8) | 41.1 (17.7) | 40.7 (16.7) |
| Donor gender: Female | 23177 (40%) | 3035 (42%) | 4457 (42%) | 1623 (45%) | 424 (43%) |
| Creatinine, Mean (SD) | 1.6 (1.70) | 1.6 (1.74) | 1.6 (1.73) | 1.6 (1.73) | 1.6 (1.81) |
| Bilirubin, Mean (SD) | 0.9 (1.04) | 0.9 (1.39) | 1.0 (1.48) | 0.9 (1.16) | 0.9 (1.45) |
| Donor Diabetes | 6311 (11%) | 808 (11%) | 1182 (11%) | 372 (11%) | 97 (10%) |
| Calculated Donor BMI, Mean (SD) | 27.6 (6.3) | 27.3 (6.2) | 27.0 (5.9) | 26.0 (5.6) | 27.4 (6.4) |
| DRI, Mean (SD) | 1.8 (0.45) | 1.8 (0.42) | 1.8 (0.45) | 1.9 (0.46) | 1.8 (0.45) |

BMI—body mass index; DRI—donor risk index; KPS—Karnofsky Performance Status; MELD—model for end-stage liver disease; SD–standard deviation.

## Sensitivity analysis and impact of socioeconomic status (SES) on waitlist removal due to death or liver transplantation

In our sensitivity analysis, Hispanics were matched 1:1 to others (Non-Hispanics) based on listing region, age, sex, BMI, MELD score, highest education, insurance and employment. By doing so, there were 18,699 patients in each group, and they were well matched (S2 Table). The analysis of matched cohorts confirmed that Hispanics had higher removal rates due to death/deterioration (Gray' test p<0.001) and lower transplant rates when compared to non-Hispanics (Fig 3A). Similarly, after liver transplant, Hispanics had better patient and graft survival than Non-Hispanic (p<0.001) (Fig 3B).

(A)

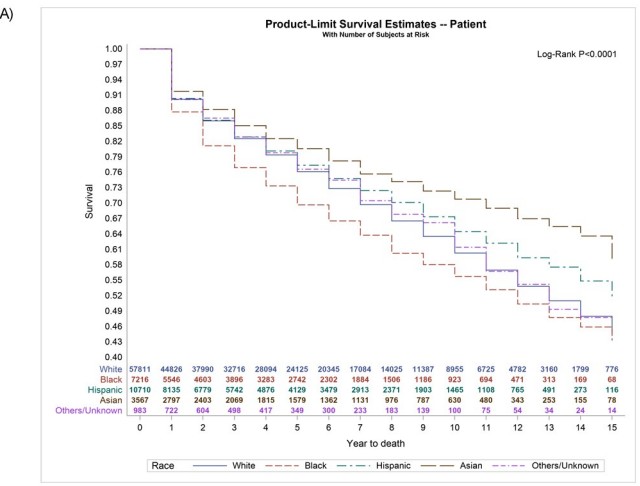

(B)

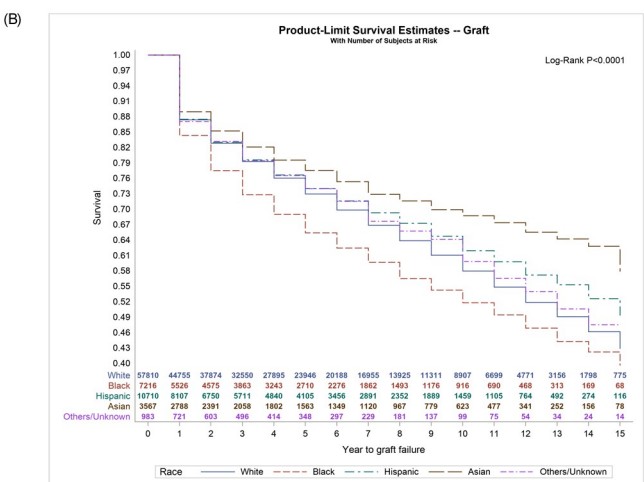

(C)

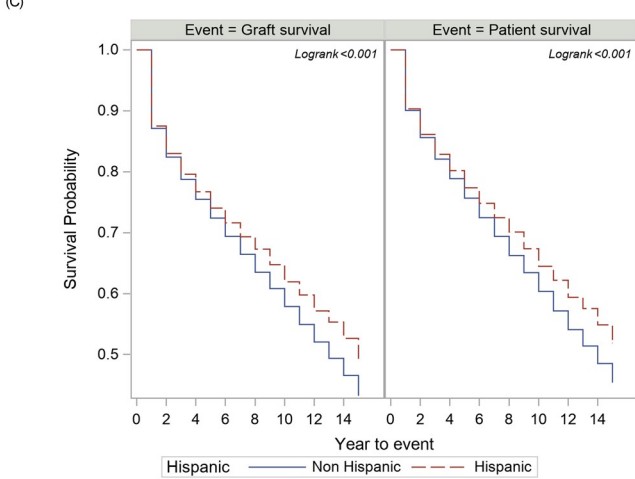

**Fig 2.** A. Post liver transplantation Kaplan Meier patient survival stratified by races (number at risk at different time points shown in inner panel); B. Post liver transplantation Kaplan Meier graft survival stratified by races (number at risk at different time points shown in inner panel); C. Post liver transplantation Kaplan Meier patient and graft survival in Hispanics when compared to others.

Table 4. Adjusted Hazard Ratios (aHR) on patient mortality and graft failure*.

| Characteristics at transplant | | Patient mortality | | | | Graft failure | | | |
|---|---|---|---|---|---|---|---|---|---|
| | | aHR | 95% CI | | P value | aHR | 95% CI | | P value |
| Age | | 1.019 | 1.017 | 1.02 | <.0001 | 1.009 | 1.008 | 1.01 | <.0001 |
| Gender (Ref = Male) | Female | 0.894 | 0.867 | 0.922 | <.0001 | 0.897 | 0.871 | 0.924 | <.0001 |
| Race (Ref = White) | Asian | 0.699 | 0.645 | 0.757 | <.0001 | 0.712 | 0.66 | 0.768 | <.0001 |
| | Black | 1.258 | 1.201 | 1.318 | <.0001 | 1.267 | 1.213 | 1.324 | <.0001 |
| | Hispanic | 0.878 | 0.839 | 0.92 | <.0001 | 0.895 | 0.857 | 0.935 | <.0001 |
| | Others/Unknown | 0.985 | 0.859 | 1.13 | 0.83 | 0.969 | 0.851 | 1.103 | 0.63 |
| Body Mass Index | | 0.994 | 0.992 | 0.997 | <.0001 | 0.996 | 0.993 | 0.998 | 0.0007 |
| MELD score | | 0.994 | 0.992 | 0.996 | <.0001 | 0.994 | 0.992 | 0.996 | <.0001 |
| Dialysis prior week to transplant (Ref = N) | Yes | 1.337 | 1.261 | 1.417 | <.0001 | 1.303 | 1.233 | 1.378 | <.0001 |
| Diabetes type 2 (Ref = N) | Yes | 1.165 | 1.122 | 1.21 | <.0001 | 1.112 | 1.072 | 1.153 | <.0001 |
| Albumin | | 0.941 | 0.922 | 0.96 | <.0001 | 0.936 | 0.918 | 0.954 | <.0001 |
| Encephalopathy is 3–4 (Ref = N) | Yes | 1.115 | 1.065 | 1.167 | <.0001 | 1.083 | 1.037 | 1.132 | 0.0004 |
| KPS categories at (Ref = 80–100) | 10–40 | 1.184 | 1.126 | 1.244 | <.0001 | 1.109 | 1.058 | 1.162 | <.0001 |
| | 50–70 | 1.11 | 1.069 | 1.152 | <.0001 | 1.076 | 1.039 | 1.114 | <.0001 |
| Donor Risk Index | | 1.274 | 1.236 | 1.313 | <.0001 | 1.346 | 1.309 | 1.385 | <.0001 |
| Medical condition (Ref = Not hospitalized) | Hospitalized not in ICU | 1.072 | 1.02 | 1.128 | 0.006 | 1.073 | 1.023 | 1.126 | 0.004 |
| | In intensive care unit | 1.289 | 1.21 | 1.372 | <.0001 | 1.304 | 1.23 | 1.384 | <.0001 |

*Clinical variables at the time of liver transplantation.

KPS—Karnofsky Performance Status; MELD—model for end-stage liver disease.

We further analyzed the data to determine the impact of SES on waitlist removal due to death or liver transplantation. We used education level, type of insurance and employment status as surrogate markers of SES for this analysis. As shown in Table 2, better SES was associated with lower waitlist mortality due to death or deterioration. When the cause-specific hazard ratios (csHR) on removal due to death or deterioration was further adjusted for SES, the higher risk for Hispanics disappeared perhaps suggesting that one of the confounders of racial disparity is SES.

## Discussion

This retrospective cohort analysis confirmed that there are significant racial disparities in waitlist removal rates due to death/deterioration, transplantation rates and post liver transplant mortality. In competing risk analysis, after adjusting for differences in clinical characteristics, Hispanics and Blacks were more likely to be removed from the waitlist removal due to death or clinical deterioration and less likely to receive liver transplantation when compared to Whites. After liver transplantation, however, Hispanics (12% higher) and Asians (30% higher) were more likely to survive as compared to whites after adjusting for other clinical risk factors.

It has been suggested that the introduction of MELD score in 2002 for liver allocation has reduced racial disparities in blacks [4, 9–11]. Our study confirmed that waitlist removal and liver transplantation rates are similar in blacks and whites, but post-LT, both graft loss and patient mortality, are ~26% higher among blacks than whites after adjusting for other covariates [4–11]. It is worth noting that blacks continue to have poor post-LT outcomes two decades after the seminal publication on this topic [5]. Further research needs to be done to identify biological and non-biological reasons for this observation in adults and children [1, 5, 6, 8]. More importantly, disparities in waitlist removal or LT rates remain a problem among

(A)

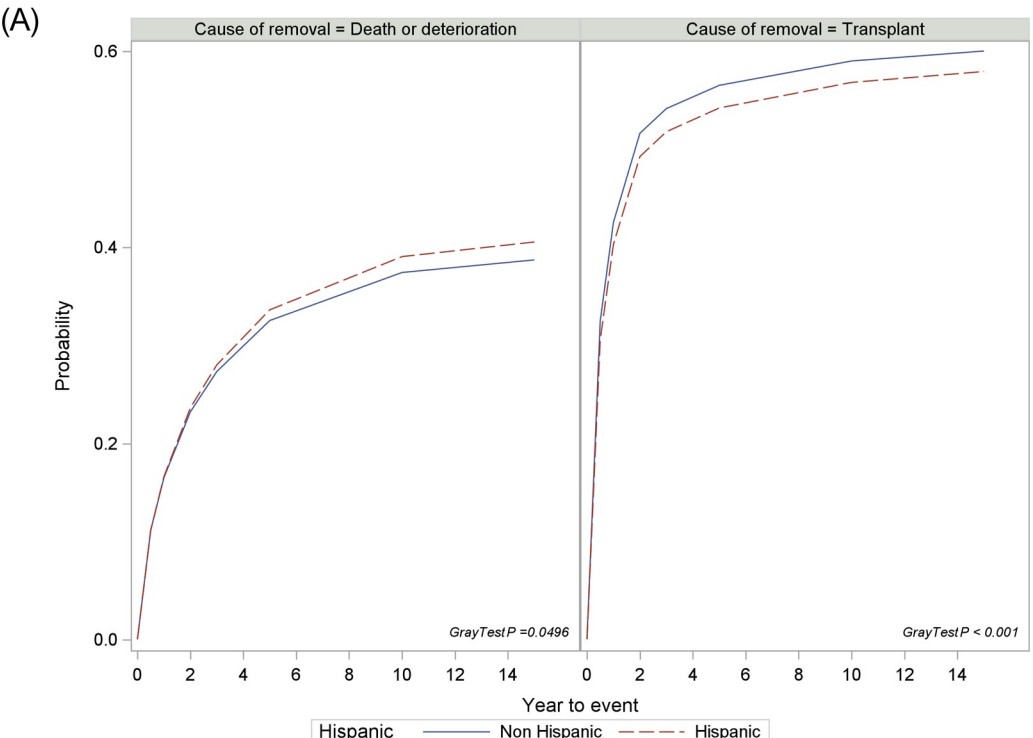

(B)

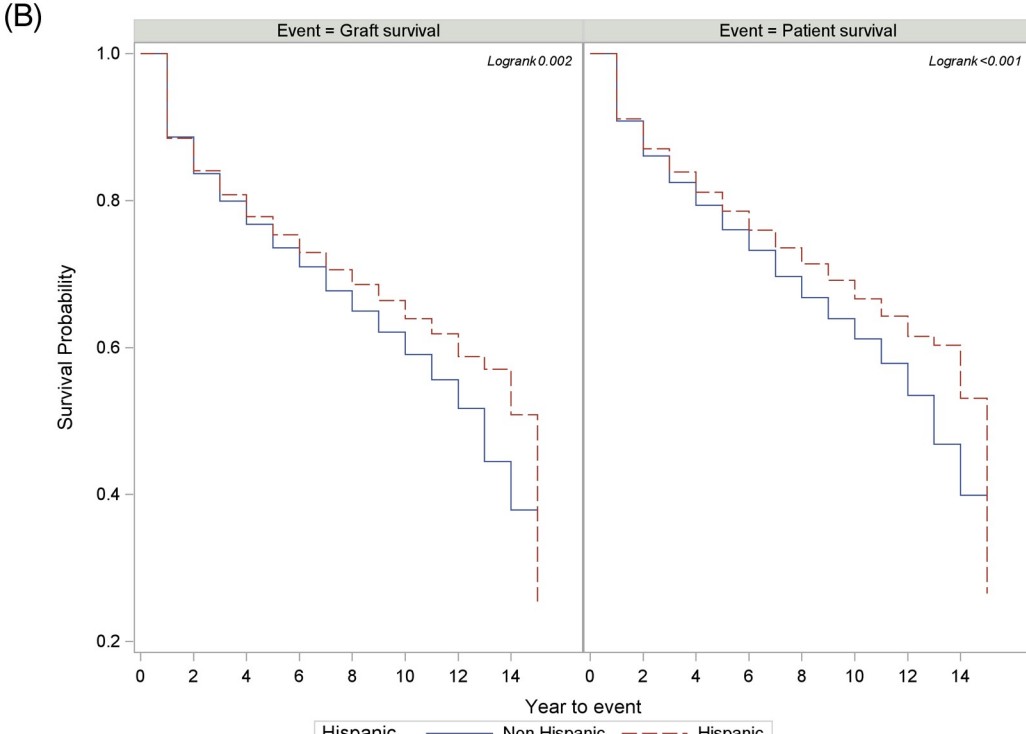

**Fig 3. A: Cumulative incidence of removal due to death or deterioration and liver transplant by Hispanic (1:1 matched cohorts); B: Kaplan Meier patient and graft post-transplant survival by Hispanics and others (1:1 matched cohorts).** The groups were matched for region, age, sex, BMI, MELD score, highest education, insurance and employment.

Hispanics, Asians and other races. Hispanics are 3.4% more likely to be removed due to death and 10% less likely to be transplanted compared to whites suggesting that the introduction of MELD for organ allocation or various other changes in organ allocation including MELD 35 share have not made a major dent in the disparities for the Hispanic population. Although our observations are not novel, this is one of the most comprehensive analyses of racial disparities in liver transplantation in the MELD era.

The racial disparities in LT could be due to overt and covert reasons and many previous reviews have addressed this in detail including interplay of multiple confounding risk factors [1–3, 7, 23]. The alleged reasons for these disparities include differences in physician access or bias, communication gaps, social class or support systems, economic status, educational level, type of insurance, cultural beliefs or perceptions, biology and so on. Socioeconomic differences have been blamed for higher waitlist removal and lower transplant rates among minorities, and if indeed these assertions were true, we would have expected similar results among all minorities including blacks. Nevertheless, when we adjusted waitlist removal for socioeconomic differences using surrogate markers such as education, insurance or employment status, the differences in waitlist removal due to death/deterioration disappeared, but disparity in transplantation remained more or less unchanged. We also do not believe higher MELD scores and lower performance status at listing among Hispanics would explain higher waitlist removal rates among Hispanics as there was concurrent lower transplantation rates, as they are not mutually exclusive, in our competing risk analyses after adjusting for the confounding risk factors. It is also unlikely that regional variations in organ availability will explain these differences in waitlist removal and transplant rates since we did our analysis after adjusting for listing region [2, 23]. It has been suggested that those living in rural areas are more likely to be disadvantaged, but Hispanics and Asians are concentrated in more urban areas [5]. Better awareness of the existing racial disparities may hopefully lead to further research in this field instead of brushing it aside as 'inevitable' inequities.

A previous study had suggested Hispanics are likely to receive organs with higher DRI (poor quality) when compared to whites [24]. Our study, however, did not find any difference in DRI between Hispanics and whites. Asians, however, received a higher proportion (33%) of poor-quality graft, defined as DRI >2, compared to others (26–29%), and yet they had the best post-LT graft and patient survival rates. Despite a higher liver disease burden, Hispanic had shown better survival. This paradox in mortality was observed in previous studies including liver diseases [25, 26]. The racial/ethnic differences in graft loss and patient mortality could be partially due to biological reasons. Other explanations include net changes in emigration where very sick Hispanics return to their country of origin and relatively less sick people with better socioeconomic status stay back and receive a LT, but this hypothesis has been challenged [27–29]. It is plausible that the improved survival in Hispanics could be due to a selection bias where those with more advanced disease were removed and relatively healthier were transplanted. If non-biological reasons are implicated for poorer outcomes in blacks, it is difficult to explain similar post-LT outcomes in Hispanics and whites as the same non-biological reasons are applicable for Hispanics.

Our study has few inherent limitations that are applicable to any studies based on retrospective cohort analyses. Our racial groups, especially Hispanics, may not be homogeneous as they may have varying geographical background (countries of origin), and differences in cultural perceptions or beliefs, educational levels, social status, communication skills, insurance or physician access. Additionally, race/ethnicity is coded based on self-reporting and moreover, UNOS data do not distinguish race from ethnicity. In addition, being an observational study, we could not identify the reasons for the observed disparities. It is also likely that racial

disparities are not fully captured by our study since we could not account for the referral bias and access to the transplant centers in a timely manner [20, 30].

The World Health Organization has defined racial disparities as "differences in health which are not only unnecessary and avoidable but, in addition, are considered unfair and unjust" [31]. The Institute of Medicine (IOM) report on unequal treatment concluded "racial and ethnic disparities in healthcare exist and, because they are associated with worse outcomes in many cases, are unacceptable" [32]. Our study suggests we have only partially succeeded in reducing racial disparities in liver transplantation and is a reminder to all of us that we have a lot more work to do at multiple levels to bring parity for all races [33].

## Supporting information

**S1 Table. Causes of death after liver transplant.**
(DOCX)

**S2 Table. Patients' characteristics comparison between Hispanics and Non-Hispanics in matched cohort.**
(DOCX)

## Author Contributions

**Conceptualization:** Paul J. Thuluvath.

**Data curation:** Talan Zhang.

**Formal analysis:** Talan Zhang.

**Methodology:** Paul J. Thuluvath.

**Supervision:** Paul J. Thuluvath.

**Validation:** Paul J. Thuluvath.

**Writing – original draft:** Paul J. Thuluvath, Waseem Amjad.

**Writing – review & editing:** Paul J. Thuluvath, Talan Zhang.

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
