## [Decision Letter · Decision Letter 0]

10 Nov 2020

PONE-D-20-33490

Liver transplant waitlist removal, transplantation rates and post-transplant survival in Hispanics

PLOS ONE

Dear Dr. Thuluvath,

Thank you for submitting your manuscript to PLOS ONE. After careful consideration, we feel that it has merit but does not fully meet PLOS ONE’s publication criteria as it currently stands. Therefore, we invite you to submit a revised version of the manuscript that addresses the points raised during the review process.

We believe that it is an interesting and well written article but it is needed to improve and to address such as the explanation regarding better graft and patient survival in Hispanic. Also please refer the comments in detail as followings.  

We look forward to receiving your revised manuscript.

Kind regards,

Yun-Wen Zheng

Academic Editor

PLOS ONE

Journal Requirements:

Reviewers' comments:

Reviewer's Responses to Questions

**Comments to the Author**

1. Is the manuscript technically sound, and do the data support the conclusions?

Reviewer #1: Yes

Reviewer #2: Yes

2. Has the statistical analysis been performed appropriately and rigorously? 

Reviewer #1: Yes

Reviewer #2: Yes

3. Have the authors made all data underlying the findings in their manuscript fully available?

Reviewer #1: Yes

Reviewer #2: Yes

4. Is the manuscript presented in an intelligible fashion and written in standard English?

Reviewer #1: Yes

Reviewer #2: Yes

5. Review Comments to the Author

Reviewer #1: This retrospective cohort analysis confirmed that there are significant racial disparities in waitlist removal rates due to death/deterioration, transplantation rates and post liver transplant

mortality, especially in Hispanic. This paper is very well written and it seems that there is no problem in general.

I have some comments.

1. The waitlist removal group is consisted only due to death or deterioration + due to liver transplantation? Or is there any cause of waitlist removal cause?

2. In the discussion p.12, the authors mentioned that when they adjusted weight list removal for socioeconomic differences using surrogate markers such as education, insurance or employment status, the

differences in waitlist removal due to death/deterioration disappeared, but disparity in transplantation remained more or less unchanged. Why the disparity in transplantation remained more or less unchanged? What causes disparity in transplantation?

Reviewer #2: Thuluvath, et al. investigated risk analysis for waitlist removal of death/deterioration and transplantation by multivariate analysis, using UNOS data of more than 150,000. They found that Hispanics had higher waitlist removal due to death/deterioration and lower transplant rates than non-Hispanics. Also, Hispanics did have better graft and patient survival than non-Hispanics. When covariables including socioeconomic status was adjusted, the higher risk in removal of death/deterioration disappeared, but not for transplantation. They discussed that these results were observational and could not find good reasons. This is interesting manuscript, but lacks data and explanations to support their theory.

Major

1. The numbers of patients are not compatible. For example, the number patients who received transplant supposed to be 154818 x 0.569 =88,091. Looking at table 3, the number of the patients were 57864+7221+10725+3573+985=80,368. Ten percent of the patient`s data are missing. These incompatibilities can lose quality of statistics of this study.

2. The authors should have better explanation regarding better graft and patient survival in Hispanics, not just to conclude with “biological reasons”. What are the biological reasons? Since Hispanics had higher MELD score with higher dialysis rates, lower KPS, etc, this population should have worse graft and patient survival. We can’t agree with the explanation for “it is plausible that ……“, since Hispanics had higher risks than other races.

3. Showing Figure 3 A and 3B without matching SEC does not make sense. It is important to show the graft and patient survival curves after matching with covariables including SEC. We think “Sensitivity analysis” paragraph and “Impact of socioeconomic status….” paragraphs should be combined to one. Also, it is meaningless to show duplicates in Table 2 and Table 5. They should only show Table 5, with reorganized paragraphs in the Result section.

4. The authors need to discuss why malignancy is the major cause of death for Hispanics.

Minor

1. We could not find the data that explains “more Asians received poor-quality graft (defined as DRI>2) than whites ……..” in Table 1.

2. Introduction and Result section, multivariate analyses for not “patient survival” and “graft survival”, we think “patient mortality” and “graft failure”.

3. What does 93.2% mean at introduction section (line 8)? The author should have more care for adding this number, or exclude this from the manuscript.

4. Page 9, in multivariate analyses for graft and patient, “gender” should be included as a risk factor.

5. Page 12, line 18, “weight list”=>”waitlist”

6. PLOS authors have the option to publish the peer review history of their article (what does this mean?). If published, this will include your full peer review and any attached files.

Reviewer #1: **Yes: **Soichiro Murata

Reviewer #2: No

---

## [Author Response · Author response to Decision Letter 0]

3 Dec 2020

PONE-D-20-33490: Response to reviewers: 

Reviewer #1: This retrospective cohort analysis confirmed that there are significant racial disparities in waitlist removal rates due to death/deterioration, transplantation rates and post liver transplant

mortality, especially in Hispanic. This paper is very well written and it seems that there is no problem in general.

I have some comments.

1. The waitlist removal group is consisted only due to death or deterioration + due to liver transplantation? Or is there any cause of waitlist removal cause?

Response: Few patients were removed from waitlist because of improved condition and other reasons as summarized below. We tried to separate removal due to improved conditions and other reasons as another competing event, but the results were similar because of smaller numbers.

 race(Race)

 White Black Hispanic Asian Others/Unknown Total

Still waiting (%) 6910

(6.3) 717

(5.5) 1954

(8.4) 545

(7.8) 155

(8.0) 10281

Died/Deteriorated/too sick (%) 29393

(26.8) 3360

(25.8) 7243

(31.2) 1696

(24.3) 555

(28.6) 42247

Transplanted (%) 63581

(58.0) 7847

(60.3) 11758

(50.6) 3912

(56.1) 1035

(53.3) 88133

Improved/other reasons (%) 9769

(8.9) 1096

(8.4) 2268

(9.8) 827

(11.9) 197

(10.1) 14157

Total 10965 1302 23223

 6980

 1942 154818

2. In the discussion p.12, the authors mentioned that when they adjusted weight list removal for socioeconomic differences using surrogate markers such as education, insurance or employment status, the differences in waitlist removal due to death/deterioration disappeared, but disparity in transplantation remained more or less unchanged. Why the disparity in transplantation remained more or less unchanged? What causes disparity in transplantation?

Response: There have been many discussions and review articles on this topic. We have cited one of our own review on this topic (#1)

1. Nguyen GC, Thuluvath PJ. Racial disparity in liver disease: Biological, cultural, or socioeconomic factors. Hepatology 2008;47(3): 1058-1066.

Reviewer #2: Thuluvath, et al. investigated risk analysis for waitlist removal of death/deterioration and transplantation by multivariate analysis, using UNOS data of more than 150,000. They found that Hispanics had higher waitlist removal due to death/deterioration and lower transplant rates than non-Hispanics. Also, Hispanics did have better graft and patient survival than non-Hispanics. When covariables including socioeconomic status was adjusted, the higher risk in removal of death/deterioration disappeared, but not for transplantation. They discussed that these results were observational and could not find good reasons. This is interesting manuscript, but lacks data and explanations to support their theory.

Major

1. The numbers of patients are not compatible. For example, the number patients who received transplant supposed to be 154818 x 0.569 =88,091. Looking at table 3, the number of the patients were 57864+7221+10725+3573+985=80,368. Ten percent of the patient`s data are missing. These incompatibilities can lose quality of statistics of this study.

Response: This was a complete oversight (wrong table included) on our side, the correct table is included in the revised manuscript. The total number is 88,133.

Variable White Black Hispanic Asian Others

 (N=63581) (N=7847) (N=11758) (N=3912) (N=1035)

Recipients’' characteristics at transplantation:

Age, Mean (SD) 54.8 (10.20) 51.6 (12.18) 53.8 (10.53) 55.2 (10.94) 52.2 (11.01)

Recipient gender: Female 20226 (32%) 3282 (42%) 4186 (36%) 1269 (32%) 423 (41%)

Recipient BMI, Mean (SD) 28.6 (5.70) 28.5 (6.18) 28.9 (5.60) 25.0 (4.43) 29.7 (5.96)

Morbidly Obese 2466 (4%) 378 (5%) 518 (4%) 21 (1%) 68 (7%)

ALBUMIN, Mean (SD) 3.1 (0.71) 2.9 (0.77) 3.0 (0.75) 3.3 (0.81) 3.0 (0.72)

TOTAL BILIRUBIN (MG/DL) , Mean (SD) 8.0 (10.45) 10.0 (11.61) 10.2 (12.47) 8.7 (12.49) 10.1 (12.28)

Serum Creatinine, Mean (SD) 1.4 (1.03) 1.5 (1.20) 1.4 (1.10) 1.2 (0.95) 1.4 (0.97)

INR, Mean (SD) 1.9 (1.31) 2.1 (2.07) 2.0 (1.28) 1.8 (1.50) 2.1 (1.20)

Meld score, Mean (SD) 21.3 (10.23) 22.9 (11.18) 23.2 (11.40) 18.9 (12.18) 23.4 (11.23)

Ascites is moderate at transplant 16883 (27%) 1754 (22%) 3256 (28%) 633 (16%) 291 (28%)

Encephalopathy is 3-4 at transplant 6593 (10%) 921 (12%) 1304 (11%) 364 (9%) 156 (15%)

KPS categories at transplant 

10 - 40 17497 (28%) 2442 (31%) 4038 (34%) 953 (24%) 354 (34%)

50 - 70 22145 (35%) 2543 (32%) 3959 (34%) 1156 (30%) 361 (35%)

80 - 100 14712 (23%) 1767 (23%) 2201 (19%) 1250 (32%) 228 (22%)

Missing 9227 (15%) 1095 (14%) 1560 (13%) 553 (14%) 92 (9%)

Child-Pugh categories at transplant 

A:5 or 6 points 6664 (12%) 1003 (14%) 1165 (11%) 1177 (33%) 115 (12%)

B:7-9 points 17372 (30%) 1765 (24%) 2846 (27%) 908 (25%) 215 (22%)

C:>9 points 33780 (58%) 4449 (62%) 6705 (63%) 1481 (42%) 655 (66%)

Medical condition 

In intensive care unit 6495 (11%) 1107 (15%) 1816 (17%) 557 (16%) 154 (16%)

Hospitalized not in ICU 9866 (17%) 1327 (18%) 2207 (21%) 465 (13%) 190 (19%)

Not hospitalized 41448 (72%) 4783 (66%) 6685 (62%) 2545 (71%) 639 (65%)

Donors' characteristics:

DONOR AGE (YRS), Mean (SD) 41.9 (16.56) 40.4 (16.13) 41.3 (16.79) 41.1 (17.69) 40.7 (16.70)

Donor gender: Female 23177 (40%) 3035 (42%) 4457 (42%) 1623 (45%) 424 (43%)

CREATININE , Mean (SD) 1.6 (1.70) 1.6 (1.74) 1.6 (1.73) 1.6 (1.73) 1.6 (1.81)

BILIRUBIN , Mean (SD) 0.9 (1.04) 0.9 (1.39) 1.0 (1.48) 0.9 (1.16) 0.9 (1.45)

Donor DIABETES 6311 (11%) 808 (11%) 1182 (11%) 372 (11%) 97 (10%)

Calculated Donor BMI, Mean (SD) 27.6 (6.31) 27.3 (6.16) 27.0 (5.94) 26.0 (5.62) 27.4 (6.43)

DRI, Mean (SD) 1.8 (0.45) 1.8 (0.42) 1.8 (0.45) 1.9 (0.46) 1.8 (0.45)

2. The authors should have better explanation regarding better graft and patient survival in Hispanics, not just to conclude with “biological reasons”. What are the biological reasons? Since Hispanics had higher MELD score with higher dialysis rates, lower KPS, etc, this population should have worse graft and patient survival. We can’t agree with the explanation for “it is plausible that ……“, since Hispanics had higher risks than other races.

Response: we have adjusted for these variables in the multivariate analysis. 

3. Showing Figure 3 A and 3B without matching SEC does not make sense. It is important to show the graft and patient survival curves after matching with covariables including SEC. We think “Sensitivity analysis” paragraph and “Impact of socioeconomic status….” paragraphs should be combined to one. Also, it is meaningless to show duplicates in Table 2 and Table 5. They should only show Table 5, with reorganized paragraphs in the Result section.

Response: We redid matching based on region, age, sex, BMI, Meld score, highest education, insurance and employment, and regenerated figure 3A and 3B, the conclusion is similar. 

We removed table 2 and only kept table 5 (table 2 in the revised MS)

Paragraphs reorganized as suggested.

4. The authors need to discuss why malignancy is the major cause of death for Hispanics.

Response: it is highest in Asians followed by whites (see supplementary table 1). 

Minor

1. We could not find the data that explains “more Asians received poor-quality graft (defined as DRI>2) than whites ……..” in Table 1.

Response: although the median DRI was similar, a higher proportion of Asians received DRI >2. We have shown that data in the result section (page 10, last sentence of 1st paragraph). 

2. Introduction and Result section, multivariate analyses for not “patient survival” and “graft survival”, we think “patient mortality” and “graft failure”.

Response: we revised the table as suggested. Thanks.

3. What does 93.2% mean at introduction section (line 8)? The author should have more care for adding this number, or exclude this from the manuscript.

Response: that was accidentally inserted and we should have done better proof reading. Thank you. 

4. Page 9, in multivariate analyses for graft and patient, “gender” should be included as a risk factor.

Response: gender added 

5. Page 12, line 18, “weight list”=>”waitlist”

Response: changed as suggested

---

## [Decision Letter · Decision Letter 1]

16 Dec 2020

Liver transplant waitlist removal, transplantation rates and post-transplant survival in Hispanics

PONE-D-20-33490R1

Dear Dr. Thuluvath,

We’re pleased to inform you that your manuscript has been judged scientifically suitable for publication and will be formally accepted for publication once it meets all outstanding technical requirements.

Kind regards,

Yun-Wen Zheng

Academic Editor

PLOS ONE

Additional Editor Comments (optional):

Reviewers' comments:

Reviewer's Responses to Questions

**Comments to the Author**

1. If the authors have adequately addressed your comments raised in a previous round of review and you feel that this manuscript is now acceptable for publication, you may indicate that here to bypass the “Comments to the Author” section, enter your conflict of interest statement in the “Confidential to Editor” section, and submit your "Accept" recommendation.

Reviewer #1: All comments have been addressed

Reviewer #2: All comments have been addressed

2. Is the manuscript technically sound, and do the data support the conclusions?

Reviewer #1: Yes

Reviewer #2: Yes

3. Has the statistical analysis been performed appropriately and rigorously? 

Reviewer #1: Yes

Reviewer #2: Yes

4. Have the authors made all data underlying the findings in their manuscript fully available?

Reviewer #1: Yes

Reviewer #2: Yes

5. Is the manuscript presented in an intelligible fashion and written in standard English?

Reviewer #1: Yes

Reviewer #2: Yes

6. Review Comments to the Author

Reviewer #1: Everything requested by the reviewer has been corrected and there is no problem. I think this manuscript is acceptable.

Reviewer #2: The paper “Liver transplant waitlist removal, trnasplantation rates and post-transplant survival in Hispanics”. is now correctly revised. The authors have answered all our comments wisely. We could not find any issues now.

7. PLOS authors have the option to publish the peer review history of their article (what does this mean?). If published, this will include your full peer review and any attached files.

Reviewer #1: No

Reviewer #2: No

---

## [Editor Report · Acceptance letter]

18 Dec 2020

PONE-D-20-33490R1 

Liver transplant waitlist removal, transplantation rates and post-transplant survival in Hispanics 

Dear Dr. Thuluvath:

I'm pleased to inform you that your manuscript has been deemed suitable for publication in PLOS ONE. Congratulations! Your manuscript is now with our production department. 

Kind regards, 

on behalf of

Dr. Yun-Wen Zheng 

Academic Editor

PLOS ONE